# Healthcare Utilization in Different Stages among Patients with Dementia: A Nationwide Population-Based Study

**DOI:** 10.3390/ijerph18115705

**Published:** 2021-05-26

**Authors:** Yu-Han Chen, Yi-Chen Lai, Yu-Cih Wu, Jun Sasaki, Kang-Ting Tsai, Chung-Han Ho

**Affiliations:** 1Department of Family Medicine, Chi-Mei Medical Center, Tainan 710, Taiwan; smallquai@gmail.com; 2Department of Emergency Medicine, An Nan Hospital, China Medical University, Tainan 709, Taiwan; johnnyliyijin@gmail.com; 3Department of Medical Research, Chi-Mei Medical Center, Tainan 710, Taiwan; cih830927@gmail.com; 4Yushokai Medical Corporation, Tokyo 105-0004, Japan; junsasakimd@gmail.com; 5Department of Geriatrics and Gerontology, Chi-Mei Medical Center, Tainan 710, Taiwan; 6Department of Senior Welfare and Services, Southern Taiwan University of Science and Technology, Tainan 710, Taiwan

**Keywords:** dementia, healthcare utilization, medical services, national health insurance

## Abstract

To evaluate the trend of healthcare utilization among patients with dementia (PwD) in different post-diagnosis periods, Taiwan’s nationwide population database was used in this study. PwD were identified on the basis of dementia diagnoses during 2002–2011. We further subdivided the cases into 10 groups from the index year to the 10th year after diagnosis. The frequency of emergency department visits and hospitalizations, the length of stay, outpatient and department visits, and the number of medications used were retrieved. The Joinpoint regression approach was used to estimate the annual percent change (APC) of healthcare utilization. The overall trend of healthcare utilization increased with the progression of dementia, with a significant APC during the first to second year after diagnosis (*p* < 0.01), except that the frequency of outpatient visits showed a decreasing trend with a significant APC from the first to fifth year. All sex- and age-stratified analyses revealed that male gender and old age contributed to greater use of healthcare services but did not change the overall trend. This study provides a better understanding of medical resource utilization across the full spectrum of dementia, which can allow policymakers, physicians, and caregivers to devise better care plans for PwD.

## 1. Introduction

Dementia is a chronic disease characterized by progressive cognitive and functional decline, which ultimately impairs the activities of daily living. It is also a major cause of disability in advanced age [1]. Previous studies showed that patients with dementia (PwD) had higher utilization of healthcare services and higher healthcare costs than those without dementia [2,3,4,5]. One global cost-of-illness study claimed that the worldwide cost of dementia increased 35% from 2010 to 2015 [6]. Because of the increasing prevalence of dementia, it has become a global public health priority.

Studies identify that the utilization of healthcare services by PwD is in large part due to increased hospitalization [7,8]; other studies report that PwD receive more hospital transition and aggressive interventions than patients without dementia near the end of life [9,10]. However, these studies did not examine medication use or distinguish the utilization of healthcare services between different stages of dementia. There are limited studies investigating healthcare utilization attributed to dementia in Asian countries. In addition, almost no study has attempted to differentiate healthcare service utilization in different post-diagnosis periods of dementia.

Taiwan is one the most rapidly aging countries and entered the stage of an aged society in 2018 [11,12], which, as defined by WHO, is when more than 14% of the population is over 65 years of age. Due to population aging, the growing number of older people with dementia poses huge challenges to public health and elderly care systems in the country. Understanding healthcare utilization throughout each patient’s clinical course is critical for patients and their families, because such knowledge can enable family caregivers and healthcare providers to make appropriate treatment plans and decisions.

Therefore, the aim of this study was to investigate the trend of healthcare utilization by patients with dementia during the post-diagnosis period. We hypothesized that utilization of healthcare resources would increase with the progression of dementia.

## 2. Materials and Methods

### 2.1. Data Source and Study Population

Taiwan’s National Health Insurance Research Database (NHIRD), an anonymized electronic record claim database, was used in this longitudinal study. Based on Taiwan’s National Health Insurance (NHI) enrollment, NHIRD covers complete inpatient and ambulatory healthcare services, including diagnosis records, reimbursement utilization, and drug prescriptions for enrolled individuals. Five diagnosis codes for admission and three diagnosis codes for outpatients are included in the NHIRD. The study was approved by the institutional review board of Chi-Mei Medical Center (approval number: 10410-E01), and informed consent was waived by the institutional review board because the NHIRD contains anonymized information only.

The study population included patients who were older than 65 years who received a diagnosis of dementia during 2002–2011. Patients were selected using the following criteria: (i) at least 3 outpatient service claims with International Classification of Diseases, Ninth Revision, Clinical Modification (ICD-9-CM) codes for dementia (290.1x–290.4x, 291.2, 292.82, 294.1x, 294.8, 331.0, 331.1x, 331.2, 331.82) within 1 year after the first dementia diagnosis code; and (ii) any single hospitalization for dementia related to the principal diagnosis codes. The criteria were also applied in our previous studies [13,14]. To ensure that the study included only patients with their first diagnosis of dementia, we excluded patients who had a history of dementia before 2002. Considering that patients with cancer had more utilization than those without, patients who were diagnosed with both dementia and cancer were excluded (ICD-9-CM cancer codes: 140–208). In addition, in order to estimate reliable trends of healthcare utilization during the study period, the minimum follow-up time was set as 2 years after the first dementia diagnosis. The follow-up time of less than 1 year among PwD was also excluded to reduce potential confounding bias. All patients were followed until death, the last follow-up date, or 31 December 2013. Death was defined by the presence of a death record in the inpatient claim dataset, ICD-9-CM diagnosis code 798 (sudden death, cause unknown) in the emergency department according to the outpatient claim dataset, or withdrawal from the insurance program without reenrollment within 180 days of the last healthcare visit. A flowchart of the study’s subject selection process is shown in Figure 1.

### 2.2. Measurements

PwD were subdivided into 10 groups from the first full year to the 10th year after diagnosis. During each observation year, the observation time could be the full year, time to death, or time until the last date of follow-up. Healthcare utilization throughout the course of dementia included (1) frequency of emergency department (ED) visits, (2) frequency of hospitalizations, (3) length of stay (LOS) during the course of dementia, (4) number of outpatient visits, (5) mean number of departments visited, and (6) mean number of medications used among PwD with chronic prescriptions, defined as chronic prescriptions ≥28 days.

### 2.3. Statistical Analysis

Frequencies with percentages and means with standard deviations (SDs)/medians with interquartile ranges (IQRs) are presented for the distributions of discrete and continuous variables, respectively. The trends for healthcare utilization were analyzed by linear regression from the index year to the 10th year after diagnosis. To describe the changes in data over the specified time interval, the annual percent change (APC) was used to estimate the trend of health utilization. APC was calculated using Joinpoint regression software [15]. In addition, stratified analysis was used to present the effects of age and sex. SAS statistical software (version 9.4; SAS Institute, Inc., Cary, NC, USA) was used to perform all statistical analyses. A *p*-value less than 0.05 was considered to be statistically significant.

## 3. Results

### 3.1. Characteristics of the Study Group

This study comprised 131,760 PwD who met the eligibility criteria, with a mean age of 78.71 ± 6.92 years. There were more female than male patients (55.37% vs. 44.63%). At the end of the study period, only 11,164 PwD completed the 10-year follow-up, and 59.22% were women. The mortality rate each year was from 11.85% to 8.54%. The baseline information of PwD during the follow-up periods is shown in Table 1. During the study period, 46.96% patients died. The median follow-up time was 4.16 years (interquartile range (IQR): 2.69 to 6.37). In addition, those patients had 1.42 ± 1.90 ED visits per year, 1.36 ± 1.41 admissions per year, LOS of 16.85 ± 23.07 days per admission per year, 34.44 ± 22.29 outpatient visits per year, and 5.92 ± 2.69 department visits. Of 68.62% patients with chronic prescriptions, the mean number of drugs for each prescription was 4.32 ± 4.43 (Table 2).

### 3.2. Utilization of Healthcare Services

Figure 2 shows that the frequency of ED visits increased with the progression of dementia. The mean number of ED visits by PwD was 2.39 times in the first year of diagnosis, then jumped up to 3.52 times in the 10th year after diagnosis. The overall trend of ED visits shows an increase from the first to the second year (*p* = 0.0026), but no further change after the second to the 10th year (*p* = 0.6249). Stratifying by sex shows that ED visits had a similar trend, with a significant difference between males and females (*p* = 0.0144). In addition, stratifying by age also shows an increasing trend of ED visits, with a significant increase for PwD aged ≥85 from the first to the second year (*p* = 0.0458) (Figure A1).

Figure 3 indicates that the frequency of hospitalizations for PwD increased with the progression of dementia. The mean admission frequency was 2.15 times in the first year of diagnosis, and rose to 3.22 times in the 10th year after diagnosis. The trend had a significant increase from the first to the second year (*p* < 0.0001), and no significant change from the second to the 10th year (*p* = 0.2807). The sex-stratified trend of hospitalization shows that males had higher hospitalization frequency than females (*p* = 0.0004). In addition, PwD aged ≥85 had more frequent hospitalizations than those aged <85 (Figure A2). However, all sex- and age-stratified trends show similar changes to the overall population.

The mean LOS among PwD with admission is plotted in Figure 4. The mean LOS was 10.60 days in the first year and rose to 11.01 in the 10th year after diagnosis. Similarly, the trend had a significant increase from the first to the second year (*p* = 0.0014), but no significant change from the second to the 10th year (*p* = 0.3432). However, the trend of LOS for males showed a significant increase from the first to the second year (*p* = 0.0017). In addition, PwD aged ≥75 also showed a significant increasing trend from the first year to the second year (*p* = 0.0062 for age 75–85; *p* = 0.0392 for age ≥85) (Figure A3).

As shown in Figure 5, PwD had a gradual decrease in outpatient visits from the first to the fifth year (*p* < 0.0001) and little change after that (*p* = 0.4123). The mean number of outpatient visits was 38.41 times in the first year of diagnosis, but decreased to 32.95 times in the 10th year (*p* = 0.0023). Stratifying by sex, the trend of outpatient visits among males and females is similar to the overall trend; females had fewer outpatient visits than males (*p* = 0.0002). In addition, the outpatient visits from the first to the fifth year among the three age groups reduced appreciably (all *p* < 0.0001), and PwD aged ≥85 had significantly fewer visits than the others (*p* = 0.0002) (Figure A4).

The mean number of departments visited decreased significantly with the progression of dementia (first year = 6.02, 10th year = 4.59; *p* = 0.0015). The number of departments visited decreased markedly from the first to the second year (*p* < 0.0001) and showed a mild gradual reduction after the second year (*p* < 0.0001). The sex-stratified trends of number of departments visited were similar to the overall trend, with a significant difference between males and females from the second to the 10th year (*p* = 0.0016) (Figure 6). In addition, Figure A5 shows that the age-stratified trends were similar to the overall and sex-stratified trends in the mean number of department visits.

Figure 7 presents the mean number of medications used for chronic prescriptions by PwD. The mean number of medications from the first year (22.89) to the 10th year (30.70) had a significant increasing trend (*p* = 0.0018). The number of chronic mediations increased markedly through the first two years (*p* = 0.0010) and more slowly up to the seventh year (*p* = 0.0177), with no significant change from year 7 to year 10 (*p* = 0.7433). Similar trends were seen for males and females, with no significant difference between genders (*p* = 0.2902). The age-stratified mean number of medications is displayed in Figure A6. PwD aged ≥85 showed an increasing trend with no significance in each cut-off year.

In summary, in the early stage of dementia, the APC of ED visits, frequency of hospitalizations, length of hospital stay, and number of medications used were significantly higher than in the advanced stage of dementia. In contrast, the frequency of outpatient visits showed a gradual decreasing trend, with a significant APC in the first 5 years.

## 4. Discussion

This study aimed to investigate the utilization of acute and subacute medical resources throughout the clinical course of dementia by using Taiwan’s population-based database. Previous research suggested that PwD have a higher frequency of hospitalizations and ED visits than patients without dementia [7,8], but there is little information on utilization of healthcare resources according to dementia severity. Studies conducted in Western countries revealed that hospitalization rates for PwD ranged from 0.37 to 1.26 per person-year [16]. Our analysis shows the frequency of admission ranged from 2.15 to 3.22 per person-year, which is higher than previous reports. In the present study, an overall increasing trend of ED visits and hospitalizations with progression of dementia was observed, that is, patients with advanced dementia had more frequent ED visits and admissions than those with an initial diagnosis of dementia. Progressive changes throughout the clinical course of dementia, such as declining cognitive function, swallowing difficulties, and development of psychological symptoms, predispose PwD to utilize more healthcare and social care services [17]. Our previous study demonstrated that the relative risk of hospitalization for PwD during the last year of life was 1.14 (0.91–1.41) [13]. People with dementia often experience longer and more frequent admissions and readmissions; however, some researchers have pointed out that these admissions might be preventable (potentially preventable hospitalizations) [17,18]. While hospitalization accounts for the majority of medical expenses near the end of life for PwD, it has been reported that there is a limited benefit for patients and their families and little satisfaction [9]. It has been suggested that optimal care for patients with advanced dementia should aim to maximize quality of life and prevent potentially inappropriate admissions and overly aggressive treatment [19,20]. Miller et al. noted that patients with advanced dementia under hospice care received less aggressive treatment and had a lower probability of hospital death [21].

To the best of our knowledge, no study has been conducted to compare different patterns of outpatient visits throughout the course of dementia. A 1-year retrospective study conducted in Taiwan showed that the mean number of outpatient visits for PwD was 36.7, which is significantly higher than for patients without dementia [2]. Their result was very similar to that of our study (33.2 to 38.41 per person-year). In the current study, in contrast to other aspects of healthcare services, the frequency of outpatient visits and number of departments visited showed a decreasing trend with the progression of dementia. Increased nursing home admissions and frequency of hospital admissions during the later stage of dementia may have contributed to this result. As dementia progresses, the severity of cognitive impairment and dependence on others for activities of daily living increase, along with behavioral and psychiatric symptoms, posing huge challenges to families and caregivers, and are positively associated with nursing home admission [22,23]. As acute illnesses and infections increase in the advanced stage of dementia, frequent ED visits and prolonged periods of hospitalization may also reduce the total number of outpatient visits. However, our results reveal that, although the frequency of outpatient visits showed a decreasing trend, the number of medications used by PwD increased significantly throughout the years after dementia diagnosis, which implies a lack of drug reconciliation and comprehensive assessment.

Consistent with other research, our analysis demonstrates a high prevalence of polypharmacy among PwD [24,25]. It is notable that the mean number of chronic medications used in the first year after diagnosis was as high as 22.89 per person-year and increased to 30.7 in the late stage of dementia, with an overall significant increasing trend. A high symptom burden in PwD may contribute to a greater need for medications; other risks such as multiple comorbidities, advanced age, depression, and a low functional status are also related to polypharmacy and potentially inappropriate medication (PIM) [26,27,28]. The number of prescribed medications is directly correlated to the risk of PIM among elderly patients with and without dementia [26,29]. Polypharmacy and PIM use among PwD has been found to be associated with multiple adverse outcomes such as long-term institutional care, functional impairment, falling down, hospitalization, and ED visits [24,30].

It was demonstrated that the prevalence of polypharmacy and PIM could be decreased by applying comprehensive geriatric assessment (CGA) among the elderly [30,31,32], and physicians with experience in geriatric, palliative, and dementia care significantly tended to discontinue medications [33]. Kase et al. also found that geriatrician management had a considerable effect on reducing the number and complexity of medications for patients with dementia [34]. Although the complexity of the disease and bothersome symptoms increase with the progression of dementia, there has been little contact with geriatricians among PwD [19]. Thus, complications would be decreased by utilizing CGA to avoid polypharmacy and PIM.

Our study identified a significant APC during the first to second year after diagnosis of dementia in all aspects of outcomes, except the frequency of outpatient visits showed a decreasing trend from the first to the fifth year. Utilization of healthcare services changed remarkably in the early stage of dementia, but not in the advanced stage (3rd to 10th year after diagnosis). One possible explanation is the diverse symptoms of dementia. PwD and their caregivers initially search for help from general practitioners and multiple physician visits rather than seeking integrated care. On the other hand, the diagnosis of dementia is often delayed. Many patients and their families are unaware of the disease until physical and behavioral symptoms occur and there are complications such as falls, fractures, or infections.

A subgroup analysis by sex and age was conducted and is presented in the figures (Figure 2, Figure 3, Figure 4, Figure 5, Figure 6 and Figure 7, Figure A1, Figure A2, Figure A3, Figure A4, Figure A5 and Figure A6). The sex-stratified trend of all aspects of healthcare services use was similar to the overall trend. However, compared to females, males had a higher frequency of ED visits, hospitalizations, and outpatient department visits, a higher number of department visits, and a longer length of stay. Stratifying by age, older PwD had more ED visits, hospitalizations, and a longer length of stay. On the contrary, younger PwD had more outpatient department visits and numbers of department visits.

Our subgroup analysis revealed that male gender and old age contributed to more use of healthcare services but did not change the overall trend, indicating that the post-diagnosis period itself is an important factor when evaluating healthcare utilization.

Most of the prior studies of dementia care have focused on the symptoms and healthcare utilization in the advanced stage or near the end of life. Our analysis points out the critical role of care planning in the early stage of dementia. This knowledge can give healthcare providers, patients, and families more realistic expectations about what they may confront as soon as the patient is diagnosed. By providing geriatrician-led integrated care, applying CGA with medication reconciliation, and using early advanced care planning (ACP) in the early stage of dementia, patients and their families may have better quality of life and satisfaction, accompanied by less burden on the healthcare system.

The strength of the present study includes the large sample size, and that this is the first study to describe changes in healthcare service utilization during the disease course of dementia by using a nationwide population-based dataset. However, limitations of this study should be noted. First, dementia diagnoses were identified from an administrative database through ICD-9-CM codes. Therefore, misclassification bias may have existed because some people with dementia received a delayed diagnosis or were undiagnosed. Second, new cancer diagnoses or other major adverse events during the follow-up evaluations were not excluded, which may affect the interpretation of the results. Third, the increased healthcare services use could also be attributed to the aging of the cohort during the study period, but not only in light of the progression of dementia. This remains a methodological limitation. Fourth, the cognitive status of PwD was not evaluated, since the disease trajectory differs from patient to patient, and the severity of dementia cannot totally be determined by the year of diagnosis. Fifth, PwDs with longer observation time had lower mortality risk. This may result from the younger diagnosed age among those patients. For addressing this issue, the related comorbidities and clinical factors should be considered in the future research. Finally, due to unrestricted access and low payments for medical services provided by the NHI, people in Taiwan tend to visit physicians more frequently than people in Western countries, and this may affect the results of healthcare utilization.

## 5. Conclusions

This is the first study to investigate healthcare utilization throughout the different periods after the diagnosis of dementia. The present study shows frequent ED visits and hospitalizations and multiple chronic prescriptions in the advanced stage of dementia. A more significant APC of medical resource utilization during the early stage (1st to 2nd year after diagnosis) than the advanced stage (3rd to 10th year after diagnosis) for patients with dementia was also addressed. These results contribute to a better understanding of the medical care burden across the full spectrum of dementia and draw attention to the impact of dementia on healthcare systems. Understanding healthcare utilization across this disease will allow policymakers, physicians, and caregivers to devise better care plans for patients with dementia. Further studies about the effects of early intervention on the utilization of healthcare resources in the first 2 years after diagnosis of dementia would be warranted.

## Figures and Tables

**Figure 1 ijerph-18-05705-f001:**
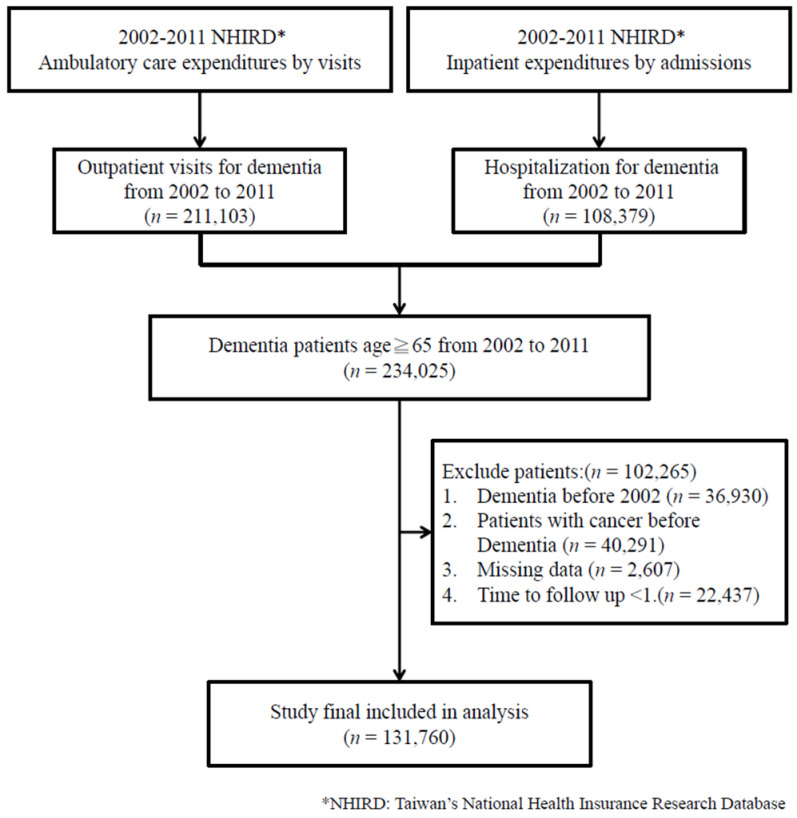
Flowchart of study subject selection.

**Figure 2 ijerph-18-05705-f002:**
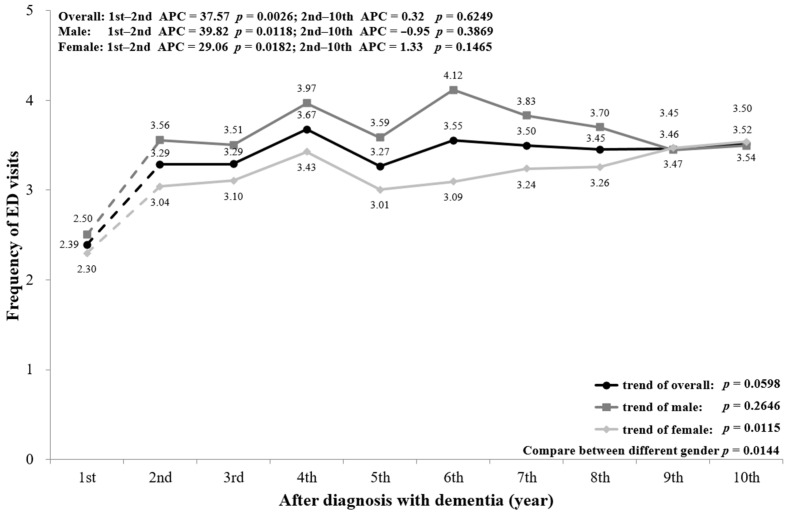
Overall and sex-stratified frequency of ED visits during the course of dementia.

**Figure 3 ijerph-18-05705-f003:**
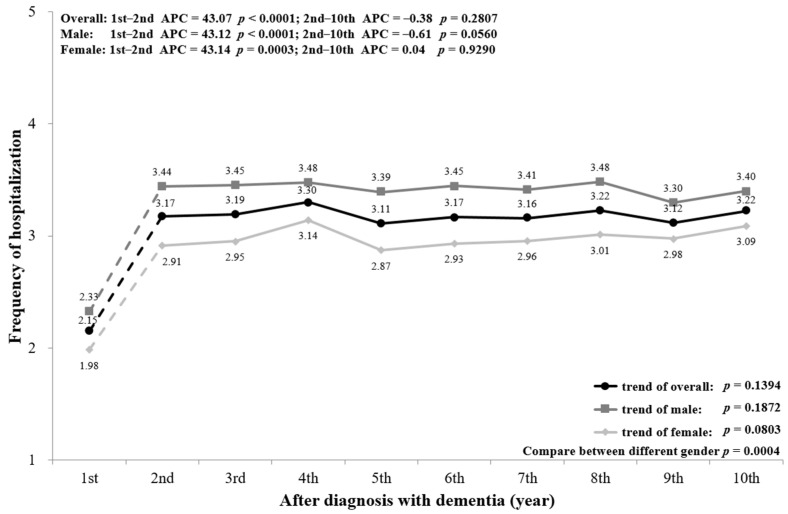
Overall and sex-stratified frequency of hospitalization during the course of dementia.

**Figure 4 ijerph-18-05705-f004:**
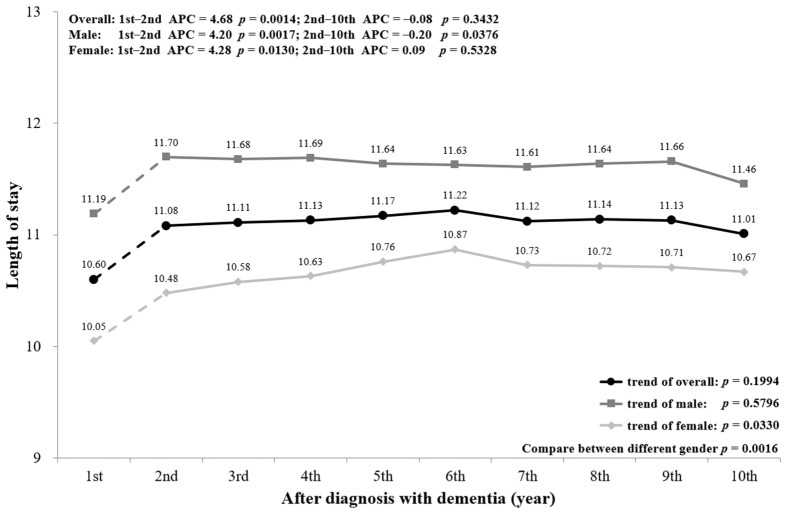
Overall and sex-stratified mean length of stay during the course of dementia.

**Figure 5 ijerph-18-05705-f005:**
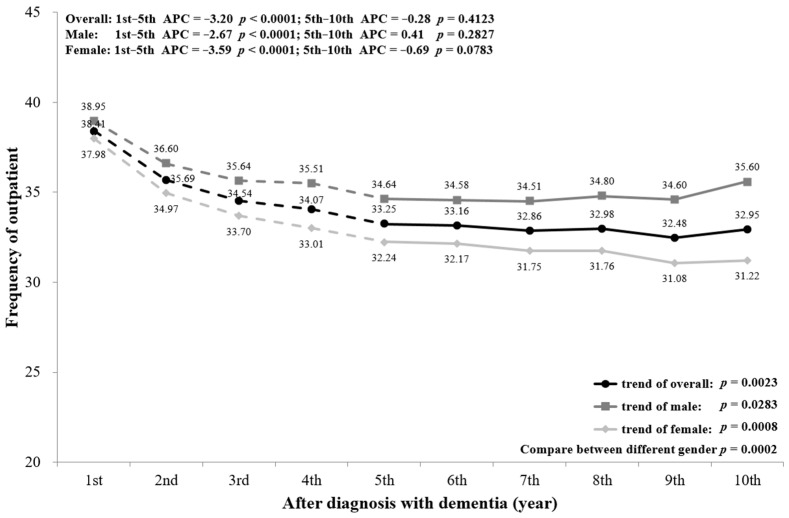
Overall and sex-stratified frequency of outpatient visits during the course of dementia.

**Figure 6 ijerph-18-05705-f006:**
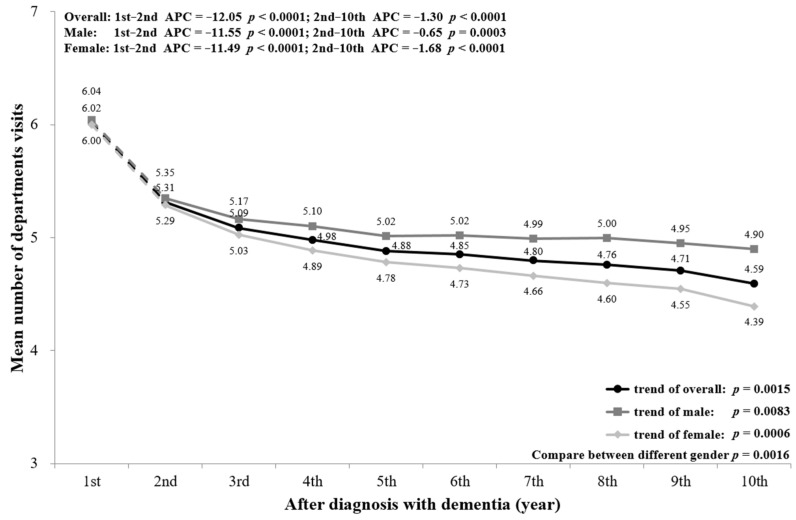
Overall and sex-stratified mean number of departments visited during the course of dementia.

**Figure 7 ijerph-18-05705-f007:**
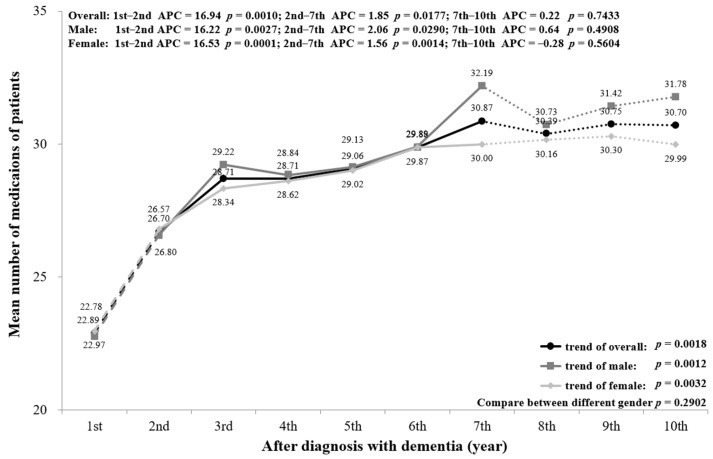
Overall and sex-stratified mean number of medications per patient.

**Table 1 ijerph-18-05705-t001:** Baseline information of patients diagnosed with dementia during follow-up period.

Year from Diagnosis	*n*	Women, *n* (%)	Deaths During Year, *n* (%)	Diagnosed Age (Mean ± SD)	Patients with Loss to Follow up
1st year	131,760	72,955 (55.37)	0	78.71 ± 6.92	0
2nd year	131,760	72,955 (55.37)	15,617 (11.85)	78.71 ± 6.92	0
3rd year	116,143	65,702 (56.57)	12,988 (11.18)	78.34 ± 6.81	12,101 (10.42)
4th year	91,054	52,241 (57.37)	10,207 (11.21)	77.85 ± 6.69	11,821 (12.98)
5th year	69,026	39,920 (57.83)	7790 (11.29)	77.36 ± 6.57	10,039 (14.54)
6th year	51,197	29,958 (58.52)	5739 (11.21)	76.90 ± 6.48	8083 (15.79)
7th year	37,375	22,090 (59.10)	4091 (10.95)	76.47 ± 6.39	7070 (18.92)
8th year	26,214	15,482 (59.06)	2792 (10.65)	76.06 ± 6.30	5672 (21.64)
9th year	17,750	10,459 (58.92)	1698 (9.57)	75.64 ± 6.23	4888 (27.54)
10th year	11,164	6611 (59.22)	953 (8.54)	75.41 ± 6.19	4134 (37.03)

**Table 2 ijerph-18-05705-t002:** Demographic characteristics and overall healthcare utilization among patients with dementia.

	Patients with Dementia (*n* = 131,760)
Age at diagnosis, *n* (%)	
65–75	71,278 (43.53)
75–85	52,326 (31.96)
≥85	8156 (4.98)
Gender, *n* (%)	
Male	58,805 (44.63)
Female	72,955 (55.37)
Death, *n* (%)	
Yes	61,875 (46.96)
No	69,885 (53.04)
Time to follow, median (Q1–Q3)	4.16 (2.69–6.37)
Variable of healthcare utilization	
Emergency frequency, *n* (%)	111,845 (84.89)
Mean ± SD	1.42 ± 1.90
Median (Q1–Q3)	0.90 (0.45–1.76)
Hospitalization frequency, *n* (%)	110,221 (83.65)
Mean ± SD	1.36 ± 1.41
Median (Q1–Q3)	0.89 (0.42–1.78)
Length of stay per admission per year	
Mean ± SD	16.85 ± 23.07
Median (Q1–Q3)	8.60 (3.09–21.06)
Outpatient frequency, *n* (%)	131,760 (100)
Mean ± SD	34.44 ± 22.29
Median (Q1–Q3)	29.73 (18.57–45.33)
Number of department visits	
Mean ± SD	5.92 ± 2.69
Median (Q1–Q3)	5.60 (4.04–7.47)
Patients with chronic prescriptions, *n* (%)	90,418 (68.62)
Number of medications for chronic prescriptions	
Mean ± SD	17.22 ± 18.37
Median (IQ25–IQ75)	10.69 (3.42–25.40)

## Data Availability

Restrictions apply to the availability of these data. Data was obtained from Taiwan’s National Health Insurance Research Database (NHIRD) and are available with the permission of NHIRD.

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
