# Peer review of "Healthcare Utilization in Different Stages among Patients with Dementia: A Nationwide Population-Based Study"

_ijerph, 2021, doi:10.3390/ijerph18115705_

Round 1

Reviewer 1 Report

I kindly ask theAuthors to see the attached file for a detailed report. 

Reviewer 2 Report

The idea to study the use of healthcare elements in different stages of dementia based on data from a nation-wide registry is fine. However, I think the authors have some more work to do before publication.

In line 49, '..the aged population...' What defines an aged population? Over 50? 60 ? 80? . Reformulate  to '..-more than 14% of the population is over xxx years.'

I have difficulties with Table 1: These are frequencies and means for patients of whom some lived through the time period studied, some died, others are in the study only during early stages of disease.  For instance, what is age? Age at diagnosis? Age at death for some, at end of study period for others, or what?   Time to death, but only for those dying during the study?  In essence, to my mind the data of Table 1 are of little interest.  I would rather like to see something like the following table:

Year from diagnosis N % women Nd- Deaths during year Mean age

N1-clinical Stage 

Mild  

N2-Clinical Stage mode-rate 

N3-Clinical Stage ad-vanced 
1. year 131760            
2. year              
3. year              
..              
..              
10. year              

If data for clinical stage of disease are not available, the table could still be presented without these data

Figures 2 to 5:  In the text commenting the data shown in the figures, I think the most interesting and most precise observations are those read from the figures, and the statistical test results should be a secondary confirmation of the statistical significance of what the graphs show. 

So for Figure 2, my preference would be the following: ..As shown in Figure 2, there is a marked increase from year 1 to year 2 (p<....), but no further change after that/from year 2 to year 10 (p=...).

Similarly in Figure 3 , again a significant increase from year 1 to year 2, and no significant change from year 2 to year 10 (p=...)

In Figure 4,  ..a gradual decrease in outpatient visits from year 1 to year 5 (p=..) and little change after that  (p=..). The number of departments visited decreased markedly from first to third year and showed a mild gradual reduction after that (p=...)

And in Figure 5, ..the number of chronic mediations increased markedly through the first two years and  more slowly after that up to year 7.

Moreover, the authors may consider the application of some variant of multiple regression to their data, to see how the outcome variables come out when controlled for age and sex distribution.

In lines 158-160: 'For the early stage of dementia ............were significantly higher in the advanced stage of dementia'  seems confusing, so reformulate and make clearer. 

I have not commented on the discussion, which should be adjusted to changes in the results chapter as suggested above.

Reviewer 3 Report

I have three four comments

1) There are a lot of typos in this manuscript. It should be carefully edited.

2)  Rationalize the use dementia cohort creation criteria. Using 3 outpatient claims within a year seems excessively narrow. Similarly, hospitalization with dementia as one of the principal diagnosis codes is not clear. How many diagnosis codes can be entered? It is not clear how diagnosis of dementia prior to 2002 was determined. In general, more clarification is needed. 

3) Please added comparable rates from other studies. The number of outpatient visits per person per year (I guess that is what frequency mean) seem very high. the number of ED visit and hospitalizations also seem high. The study can present numbers for dementia patients in other countries or for other diagnosis.

4) There are several unsupported conclusions. For example, as noted at the end of abstract, the study do not provide any evidence that early integrated care may optimize utilization. Such conjecture should be removed.

Round 2

Reviewer 1 Report

I ackowledge that the Authors substantially improved the overall quality of he manuscript, and recognized the presence of additional limitations of the study.

Author Response

Thanks for the reviewer's comment.

Reviewer 2 Report

The revised manuscript has been improved with respect to language, which is ok now.  My previous comments and suggestions have largely been followed.

However, I see some mistakes or misinterpretations or less meaningful statements.

  1. In the new Table 1, which I asked for, mortality in 1st year is 0, contrasting with over 11 % in each following year. I realize that this is because patients with less than 2 years of follow-up were excluded from the material.  But it also sheds some lights on the other data recorded for the 1st year. Going back to the 'Measurements' section, I note the definition of index year, which seems to be identical with what is defined as 1st year, as 'the year in which patients received the diagnosis of dementia for the first time'.  This obviously means that 1st year is the calendar year , not the first full year of observation of each patient. This explains the striking differences in frequency of events between 1st year and 2nd and following years, since 1st year observations are only from variable fractions of the year, whereas observations from following years covers the full year for each patient (except for those lost by death during the year).  This makes it not meaningful to include 1st year observations in the trend analyses of figures 2-7 and invalidates their statements of significantly increased frequency between 1st and second years, and also invalidates the statements of significant overall frequency of events.   I assume this is also the reason why the first segment of the trend lines are shown as broken lines, which is not otherwise explained.  It is also not explained why the curve lines in Figure 5 are broken lines  up to 5th year.
  2. In the new Table 1, they also show a striking decreasing mortality rate from 1st to 10th year, and a parallel increase in mean age. As this is a relationship between mortality and age that is opposite of the usual, it should at least be commented and explained.
  3. Line 118... states the median survival time from diagnosis to death from dementia for those patients that died during the observation period.  As this is a selection of patients who died early, The statement of median or mean survival time is of little interest.

The points referred to above are so grave that I cannot recommend this for publication. The material still contains information of value, but I believe the whole work should be redone with competent statistical assistance.

Reviewer 3 Report

The authors addessed all my comments adequeately.

Author Response

Thanks for the reviewer’s comment.